

# The effect of flywheel complex training with eccentric-overload on muscular adaptation in elite female volleyball players

Jiaoqin Wang[1,2,*], Qiang Zhang[1,*], Wenhui Chen[3], Honghao Fu[4], Ming Zhang[2] and Yongzhao Fan[5]

[1] Capital University of Physical Education and Sports, Beijing, China
[2] Beijing Sport University, Beijing, China
[3] Shen Zhen Harbor School, Shenzhen, China
[4] Huazhong University of Science and Technology, Wuhan, China
[5] Department of Physical Education, Henan Normal University, Xinxiang, China
[*] These authors contributed equally to this work.

## ABSTRACT

This study aimed to compare the effects of 8 weeks (24 sessions) between flywheel complex training with eccentric overload and traditional complex training of well-trained volleyball players on muscle adaptation, including hypertrophy, strength, and power variables. Fourteen athletes were recruited and randomly divided into the flywheel complex training with an eccentric-overload group (FCTEO, $n = 7$) and the control group (the traditional complex training group, TCT, $n = 7$). Participants performed half-squats using a flywheel device or Smith machine and drop jumps, with three sets of eight repetitions and three sets of 12 repetitions, respectively. The variables assessed included the muscle thickness at the proximal, mid, and distal sections of the quadriceps femoris, maximal half-squats strength (1RM-SS), squat jump (SJ), countermovement jump (CMJ), and three-step approach jump (AJ). In addition, a two-way repeated ANOVA analysis was used to find differences between the two groups and between the two testing times (pre-test $vs.$ post-test). The indicators of the FCTEO group showed a significantly better improvement ($p < 0.05$) in CMJ (height: ES = 0.648, peak power: ES = 0.750), AJ (height: ES = 0.537, peak power: ES = 0.441), 1RM-SS (ES = 0.671) compared to the TCT group and the muscle thicknes at the mid of the quadriceps femoris (ES = 0.504) after FCTEO training. Since volleyball requires lower limb strength and explosive effort during repeated jumps and spiking, these results suggest that FCTEO affects muscular adaptation in a way that improves performance in well-trained female volleyball players.

## INTRODUCTION

In competitive team sports like basketball, volleyball, and American football, muscle strength and power are essential, forming the foundation for performance-determining

Corresponding authors
Ming Zhang, 2147@bsu.edu.cn
Yongzhao Fan,
fanyongzhao@cupes.edu.cn

activities such as jumping, running, and hitting (*Comfort et al., 2014*; *Cronin & Hansen, 2005*; *Newton & Kraemer, 1994*; *Sheppard et al., 2008*; *Ziv & Lidor, 2009*). *Sheppard et al. (2008)* emphasized that the critical role of jumping in volleyball, contributing to offense (greater height to hit over the block/greater angle of attack) and also on defense (achieving a higher block position) (*Sheppard, Borgeaud & Strugnel, 2008*; *Sheppard et al., 2008*; *Sheppard et al., 2007*), while the strength and power in an athlete's lower limbs are major contributors to their jumping capacity (*Young, Wilson & Byrne, 1999*).

Muscle strength can be influenced by various factors, including muscle moment arm, muscle size, and activation (*Vigotsky, Contreras & Beardsley, 2015*). Muscle size is particularly significant due to its high plasticity (*Flück & Hoppeler, 2003*) and the positive correlation between muscle cross-sectional area (CSA) and strength, greater CSAs often correlate with higher strength capacities (*Maughan & Nimmo, 1984*). Skeletal muscle hypertrophy in resistance training is typically seen as an increase in muscle size, such as muscle thickness and CSA across the entire muscle tissue (*Russell, Motlagh & Ashley, 2000*). There is a robust correlation ($r = 0.5$–$0.6$) between changes in strength and hypertrophy following resistance training (*Akagi et al., 2020*; *Erskine, Fletcher & Folland, 2014*). Power is calculated as strength multiplied by velocity (*Cardinale, Newton & Nosaka, 2011*), and factors that influence muscular strength also affect power generation capabilities (*McArdle, Katch & Katch, 2006*). There is a clear relationship between muscle hypertrophy, strength, and power.

Coaches and researchers have explored various methods to improve hypertrophy, maximal strength, and power. Methods to develop power include developing maximal strength, force rate, ballistic movements, plyometrics, and technical exercises (*Bompa, 1999*; *Kraemer & Newton, 1994*; *Zatsiorsky, Kraemer & Fry, 2020*). Athletes must enhance both maximal muscle strength and movement speed to achieve great power. Complex training (CT), a form of combination training, Several sets of high-load (*e.g.*, back squat) exercises completed before the execution of several sets of low-load, higher-velocity (*e.g.*, vertical jump) exercises within the same session (*Chu, 1996*; *Ebben, 2002*; *Ebben & Blackard, 1997*), has been shown to increase both strength and power (*Carter & Greenwood, 2014*; *Pagaduan & Pojskic, 2020*). However, the traditional resistance training (TRT) component of traditional complex training (TCT) primarily utilizes weight plates, and the resistance exercises involve sequences of concentric and eccentric actions. During these exercises, an individual's ability to perform a maximal concentric-eccentric cycle is constrained by the force-velocity relationship (*Hill, 1938*). While performing the concentric-eccentric cycle during TRT, muscles can achieve greater absolute forces during eccentric actions than concentric ones (*Alkner et al., 2003*). However, resistance exercises typically provide only 40% to 50% of the maximal eccentric load during the eccentric phase (*Dudley et al., 1991*). This results in the eccentric phase being considerably under-loaded, as it is limited by the load used during the concentric phase (*Dudley et al., 1991*). Furthermore, *Norrbrand, Pozzo & Tesch (2010)* observed that maximal muscle activation during TRT occurs at the contraction failure or "sticking point", leading to a decrease in maximal forces from the first repetition and a decline in force throughout the set (*Norrbrand, Pozzo & Tesch, 2010*). Therefore, the development of new training techniques

is essential to address the limitations of TRT and improve muscle hypertrophy, strength and power (*Perez-Gomez & Calbet, 2013*).

Inertial-resistance training, especially when using flywheel devices, has emerged as an alternative to TRT, potentially compensating for TRT's limitations in TCT. Flywheel devices offer unlimited resistance throughout the entire range of motion (*Norrbrand et al., 2008*; *Norrbrand, Pozzo & Tesch, 2010*), optimizing muscle loading at each joint angle (*Tesch, Fernandez-Gonzalo & Lundberg, 2017*). Inertial loading ensures accommodated resistance, allowing maximal forces to be generated from the first to the last repetition of the set (*Norrbrand, 2008*). Moreover, properly conducted flywheel exercises may offer a safer and more effective eccentric phase than TRT (*Maroto-Izquierdo et al., 2017b*; *Raya-González, Castillo & Beato, 2021*; *Raya-González et al.*; *Tesch, Fernandez-Gonzalo & Lundberg, 2017*), leading to improved physical capacity and athletic performance-related adaptations (*Beato & Dello Iacono, 2020*; *Beato et al., 2020*; *Bright et al., 2023*; *De Keijzer, Gonzalez & Beato, 2022*; *Liu et al., 2020*; *Petré, Wernstål & Mattsson, 2018*). Previous studies have shown that high-load intensity eccentric training is more effective in developing muscle strength and hypertrophy compared to low-load eccentric training (*Dudley et al., 1991*; *English et al., 2014*; *Hakkinen, 1981*; *Norrbrand et al., 2008*; *Norrbrand, Pozzo & Tesch, 2010*). Furthermore, various studies have demonstrated the superiority of flywheel training over TRT in increasing muscle strength, vertical jumping ability (*Maroto-Izquierdo et al., 2017b*; *Puustinen et al., 2023*), and improving post-activation performance enhancement (PAPE) (*Norrbrand, Pozzo & Tesch, 2010*). Overall, flywheel exercises are considered valid and effective training methods for improving muscular adaptation.

Owing to the limited research in flywheel complex training, this study aimed to compare the effects of an eight-week periodized strength/power training program between flywheel complex training with eccentric overload (FCTEO) and TCT on muscle thickness, power, and strength in well-trained volleyball players.

## METHODS

### Experimental approach to the problem

A randomized two-group design with repeated measures was utilized. The sample size was determined using G*Power Software (*Faul et al., 2007*), with a power of $(1 - \beta)$ .90, an alpha error of .05, and an effect size of .58 based on a previous study examining the benefits of Flywheel Resistance Training (FRT) in team sport participants (*Seynnes, De Boer & Narici, 2007*). Consequently, a minimum of 12 participants was required. Fourteen participants ($n = 14$) were then randomly assigned either to the FCTEO group, which performed RT with a flywheel device, or to the control group (the Traditional Complex Training group, TCT group), which performed RT with a Smith machine. There was no turnover of players due to closed training before the tournament. All initially enrolled subjects completed the study. All other training factors, such as specific training, rest, movement tempo, training attire, and diet, remained constant. The training sessions were performed three times per week for eight weeks. Muscle thickness, strength, and power were measured before and after the program (Fig. 1).
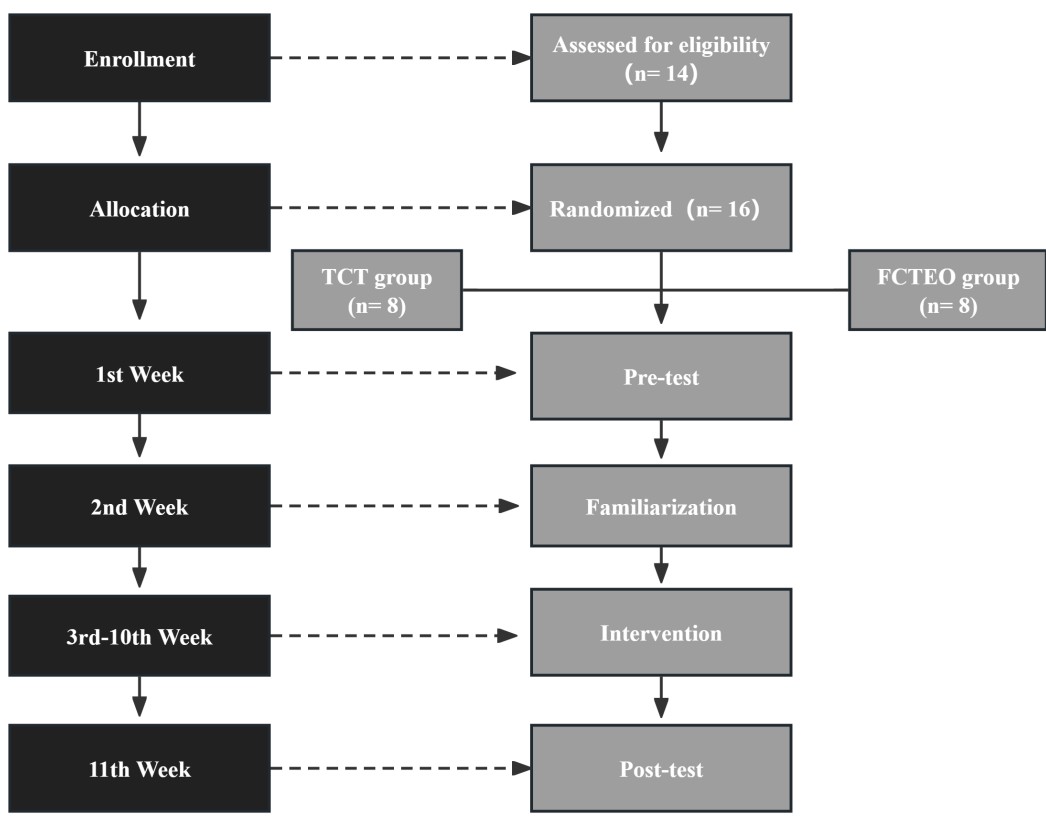

**Figure 1** Design of the entire study, including all points of measurement.

**Table 1** Information on the groups' anthropometry.

|  | FCTEO | TCT | P-value |
|---|---|---|---|
| Age (years) | 21.0 ± 2.3 | 21.1 ± 1.2 | .85 |
| Height (cm) | 179.8 ± 5.9 | 179.1 ± 4.9 | .82 |
| Weight (kg) | 67.4 ± 8.8 | 67.5 ± 6.6 | .97 |

## Subjects

Fourteen women from a high-level, Division I volleyball team were selected for the study. In the preceding three months, they engaged in volleyball practice at least four times weekly, Including RT and jumping exercises. The ages, heights, and weights of the participants are listed in Table 1. Each participant provided written consent to participate in this research, which was approved by the Ethics Council of the Capital University of Physical Education and Sports (2021A39).

Testing was conducted before (pre) and after (post) 8 weeks of training. The research was carried out during the off-season, as part of the annual periodization. The primary exclusion criteria included a lower limb joint injury within 6 months before the study and/or a severe lower limb muscle injury (strains lasting more than 27 days) within 2 months prior to the study (*Maroto-Izquierdo, García-López & De Paz, 2017a*). Athletes

who sustained an injury during the experimental phase were excluded, though no participant was excluded from the research. Additionally, throughout the study, all athletes followed consistent dietary regimens tailored to their body weight and activity needs, which were recorded using 24-hour quantitative food frequency questionnaire (*Food and Agriculture Organization of the United Nations (FAO), 2018*). Athletes were encouraged to monitor their sleep, consistently achieving the recommended 8 h per night. Alcohol consumption was minimal to nonexistent.

## Experimental procedures

Prior to beginning intervention training, all athletes participated in four testing sessions to assess anthropometry, muscle thickness (MT), strength, and vertical jumping performance. These testing sessions were scheduled on Tuesday, Wednesday, and the weekend during the initial week. Specifically, on Monday, participants were dedicated to familiarizing themselves with each testing technique. Before any testing, participants completed a 15-minute general warm-up consisting of dynamic stretching, cycling, and rowing, as well as submaximal familiarization exercises for the assessment exercises. Basic anthropometric measurements, including the subjects' weight, age, height, and training history, were taken on Tuesday. Subsequently, ultrasound imaging was utilized to determine MT. On Wednesday, one maximal repetition (1RM) for the parallel half-squats was measured. Specifically, the subjects underwent a progressive resistance loading test for the 1RM parallel half-squat (top of the thigh parallel to the ground). We instructed the participants to descend and rise without halting until the knee and hip joints were fully extended. An optical encoder (ChronoJump Co., Barcelona, Spain) with an accuracy of one mm and a sampling rate of 1,000 Hz was mounted to the barbell to measure displacement during both the concentric and eccentric phases. To ensure uniformity across trials and between sessions, the knee flexion angle was evaluated *via* video analysis (Hudl Technique App; Agile Sports Technologies, Lincoln, NE, USA). Participants began the parallel half-squat at 50% of their 1RM, with the increments for each load determined according to the technique specified by *Brown & Weir (2001)*. On the weekend, muscular power tests, including SJ, CMJ, and the three-step AJ, were measured. The participants completed three maximal SJs, CMJs, and three-step approach AJs, with two to three minutes of rest in between.

After the aforementioned tests were completed, athletes were thoroughly familiarized with the flywheel device and exercise technique over the course of a week, with three sessions scheduled on Wednesday, Friday, and Sunday. During the familiarization period, the FCTEO group's subjects completed a progressive loading test in the parallel half-squat utilizing a flywheel device (Desmetec Full 11), beginning with a 0.12 kg m2 moment of inertia. We increased the moment of inertia for the flywheel until the velocity of the flywheel was similar to that caused by the barbell parallel half-squat using 80% of 1RM intensity. Each repetition consisted of a maximal concentric action accelerating the wheel, followed by an eccentric action decelerating the wheel to a stop at approximately 90° knee flexion (*Fernandez-Gonzalo et al., 2014*). In the meantime, the TCT group's subjects completed the parallel barbell half-squat utilizing a Smith machine. A washout interval

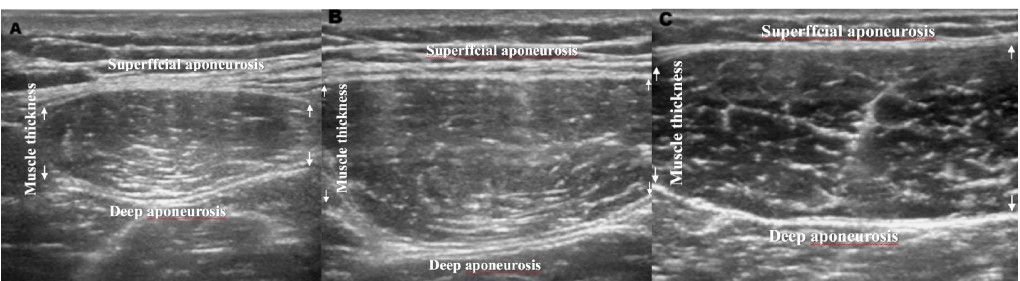

**Figure 2** Images for quadriceps femoris.

of 48 to 72 h was allowed between the familiarization period and the formal training. Participants conducted the formal training sessions (FCTEO or TCT) on Wednesday, Friday, and sunday of each week following the familiarization period.

## Muscle thickness assessment

Athletes' MT measurements were taken. Participants were instructed to fast for 12 h prior to the test, abstain from consuming alcohol for 24 h, refrain from vigorous exercise for 24 h, and urinate immediately before the exam. Ultrasonic imaging was utilized to determine MT. Ultrasonography has been used in various investigations to evaluate hypertrophic changes (*Miyatani et al., 2004*; *Pretorius & Keating, 2008*), and has been proven to be a good predictor of gross muscular hypertrophy in these muscles (*Abe et al., 2000*; *Nogueira et al., 2009*). It has been stated that the reliability and validity of ultrasonography in identifying MT are very good when compared to the "gold standard" of magnetic resonance imaging, with mean intraclass correlation coefficients (ICC) of 0.998 and 0.999 for reliability and validity, respectively (*Reeves, Maganaris & Narici, 2004*). Testing was performed by an experienced technician using a B-mode ultrasound machine (ECO3; Chison Medical Imaging, Ltd, Jiangsu, China). The technician, who was not blinded to group assignment, applied a water-soluble transmission gel (Aquasonic 100 ultrasound transmission gel) to each measurement site and then placed a 5 to 10 MHz ultrasound probe perpendicular to the tissue surface without pressing the skin. Images for MT measures were captured at distances of 25%, 50%, and 75% between the greater trochanter of the quadriceps femoris and the lateral condyle of the femur. MT was measured as the mean distance between the superffcial and deep aponeurosis at both image extremities (*Oliveira, Carneiro & Oliveira, 2016*) (Fig. 2). To maintain consistency from session to session, sites were measured with a vinyl measuring tape and then marked with a felt pen. During measurements of the quadriceps, participants stood with their legs stretched and relaxed. To prevent training-induced muscle swelling from obscuring the results, pictures were taken 48–72 h before the commencement of the study and following the final training session (*Ogasawara et al., 2012*).

## Vertical jump testing

The participants performed three maximal-effort jumps and submaximal familiarization trials with the assessment jumps, which included the squat jump (SJ), the counter-movement jump (CMJ), and the three-step approach jump (AJ). In the SJ, players were required to descend into the starting position, achieving a knee angle of 90 degrees, and then stabilize for two to three seconds before performing the jump without a counter-movement. Throughout the jump, participants were instructed to keep their palms on their ipsilateral hip. For the CMJ, participants executed a rapid countermovement to attain a 90-degree knee angle, followed by an immediate powerful extension through the hips, knees, and ankles to leap as high as possible (*Kozinc et al., 2022*). The AJ entailed a three-step approach and countermovement before the jump (*Sattler et al., 2012*). In all jump tests, participants were required to take off from the contact jump mat and avoid bending their knees in the air (*Markovic et al., 2004*). Three trails completed for each test, with intervals of two to three minutes between trials. The performance of all jump tests was assessed using a contact platform (SmartJump; Fusion Sport, Coopers Plains, Australia). The flight time $[t]$—the time interval between take-off and landing—was used to estimate the rise height of the of the body's center of gravity during the jumps (*Loturco et al., 2017*). The height jumped was calculated using the formula: Height $= 0.5 * 9.81 *$ ((flight time/2)$^2$) (*Hughes, Massiah & Clarke, 2016*), Peak power was assessed using the equation (61.9 * jump height [cm] + 36.0 * body mass [kg] $-$ 1822) (*Sayers et al., 1999*). The highest jump height were recorded for calculate the peak power.

## Maximal half-squat strength testing

After consuming breakfast and at least 750 ml of liquid, the athletes assembled at 9:00 AM, having been instructed to obtain a minimum of 8 h of sleep the previous night. To quantify maximal strength, athletes performed a standard warm-up followed by submaximal familiarization trials with the assessment activities. The half-squat exercise, known for its well-established effectiveness, was chosen for 1RM (one-repetition maximum) testing. Under the guidance of a professional strength and conditioning coach, the athletes executed controlled half-squats to just below parallel. Following the protocol described *Brown & Weir (2001)*, athletes performed sets of five reps at 50% 1RM, three reps at 60% 1RM, and two reps at 80% 1RM, progressing to single repetitions at 90%, 95%, and 100% 1RM, based on their individual records of historical performance. If successful at the 1 $\times$ 100% 1RM lift, the athlete continued to increase the weight by 2.5 kg per attempt until failure. The maximal load lifted was recorded as the athlete's 1RM. A 5-minute rest period was provided between each lifting attempt for passive recovery.

## Training programs

The participants engaged in a complex training regimen for the lower limbs that included half-squats and drop jumps (Fig. 3). This eight-week training program involved three weekly sessions, each separated by at least 48 h (*Norrbrand et al., 2008*). Each session comprised three sets of eight repetitions of half-squats and three sets of twelve repetitions of drop jumps. Workouts began with a 6-minute cycling warm-up, followed by two sets

**Figure 3** Experimental design.

of 5 bodyweight drop jumps and two sets of 10 non-maximal repetitions of specific half-squat exercises. The flywheel device (Desmetec Full 11) used by the FCTEO group was set to an intensity that provided a mean velocity similar to the velocity achieved with 80% of the 1RM half-squat, as established in previous studies (*Carroll et al., 2019*; *Mike et al., 2017*; *Pereira et al., 2016*; *Schoenfeld et al., 2016*). This approach was based on findings that training with high loads (80% 1RM) can be more effective in improving muscle strength and recruitment than low-load training (*Schoenfeld et al., 2016*). The FCTEO regimen involved extending and flexing the knees and hips to accelerate and decelerate the flywheels (*Fernandez-Gonzalo et al., 2014*). In brief, each repetition required the participants to rotate the wheels with a maximum concentric push, extending from 90° of knee flexion to near full extension. Participants were instructed to exert maximum effort throughout the entire concentric phase. At the end of this movement, the inertial forces caused the flywheel strap to coil back, initiating the eccentric phase. During the first third of the eccentric action, participants were advised to resist gently, then apply maximal braking power to halt the movement at approximately 90° knee flexion (*Fernandez-Gonzalo et al., 2014*), thus creating eccentric overload (*Norrbrand, Pozzo & Tesch, 2010*; *Romero-Rodriguez, Gual & Tesch, 2011*; *Tesch et al., 2004*). The TCT group performed half-squats on a Smith machine, with both groups maintaining the same training volume at a load equivalent to 80% 1RM (*Hanson, Leigh & Mynark, 2007*). After the resistance training, participants performed drop jumps, starting with a box height of 45 cm in the first week and increasing to 60 cm by the eighth week. Participants were required to provide a rating of perceived exertion on a 10-point scale after each set of half-squats, whether performed with a flywheel or a barbell. The number of sets (*Timon et al., 2019*) and repetitions (*Güllich & Schmidtbleicher, 1996*; *Mihalik et al., 2008*) were determined based on prior research. Rest intervals between sets lasted 60 s, and those between exercises lasted 2 min (*Fleck & Kraemer*).

## Statistical analysis

Statistical analysis was conducted using SPSS version 20.0 (SPSS Inc., IBM, Armonk, NY, USA). Data presented in tables and graphs are depicted as mean ± standard deviation.

**Table 2  Muscle thickness and strength and power tests before the training period.**

| Variables | FCTEO | TCT | P |
|---|---|---|---|
| QF25 (mm) | 58.9 ± 2.1 | 58.6 ± 7.9 | .92 |
| QF50 (mm) | 46.6 ± 4.6 | 49.0 ± 8.0 | .51 |
| QF75 (mm) | 37.0 ± 3.0 | 39.9 ± 8.2 | .39 |
| 1RM-SS (kg) | 117.1 ± 11.5 | 115.0 ± 11.9 | .74 |
| SJ height (cm) | 28.2 ± 2.7 | 28.4 ± 3.2 | .92 |
| SJ peak power (w) | 2726.4 ± 305.1 | 2969.3 ± 214.3 | .11 |
| CMJ height (cm) | 35.7 ± 2.7 | 35.5 ± 3.8 | .09 |
| CMJ peak power (w) | 3128.9 ± 223.4 | 3152.0 ± 161.8 | .83 |
| AJ height (cm) | 43.8 ± 2.3 | 41.5 ± 3.7 | .18 |
| AJ peak power (w) | 3645.6 ± 258.2 | 3582.0 ± 167.5 | .60 |

The Shapiro–Wilk, Levene, and Mauchly tests were employed to assess the variances of the sample data for normality, homogeneity, and sphericity, respectively, respectively. For pre-test between-group comparisons, a univariate analysis of variance (ANOVA) was utilized. The effects of the experimental intervention were evaluated using a two-way repeated-measures ANOVA, with factors being group (FCTEO *vs.* TCT) and time (pre-test *vs.* post-test at 8 weeks). In cases where an interaction or main effect was found to be statistically significant, *post hoc* comparisons were performed using the Bonferroni correction to identify mean differences. Statistical significance was set at $P \leq 0.05$. The effect size (ES) of the training was calculated using partial eta squared, as suggested by *Cohen (1965)*. Cohen classified ES values as small (0.01), medium (0.06), and large (0.14) to represent modest, moderate, and substantial effects, respectively.

## RESULTS

Participants from both the FCTEO and TCT groups successfully completed all training sessions. Following the training sets, the levels of perceived effort reported by the FCTEO and TCT groups were comparable. Prior to the initiation of the training session, no significant differences were observed between individuals in the FCTEO and TCT groups in terms of measurable variables, as shown in Table 2. According to diary entries, all experimental subjects, with the exception of one, adhered to the prescribed FCTEO and TCT training programs. No adverse effects, other than delayed muscle soreness, were observed in either the FCTEO or TCT groups during the training intervention. Additionally, there was no significant difference in the rating of perceived exertion between the FCTEO and TCT groups throughout the training intervention (FCTEO group: 7.1 ± 1.6, TCT group: 7.3 ± 2.4, $P > .05$).

### Hypertrophy

The results of muscle thickness (MT) of the quadriceps femoris (QF) are presented in Fig. 4. There was no significant interaction effect between group and time for MT at 25% of the QF length (QF25) (F_group × time = 0.015, $P = 0.0908$, ES = 0.002). Similarly, the main effects of time (F_time = 0.038, $P = 0.852$, ES = 0.006) and group (F_group

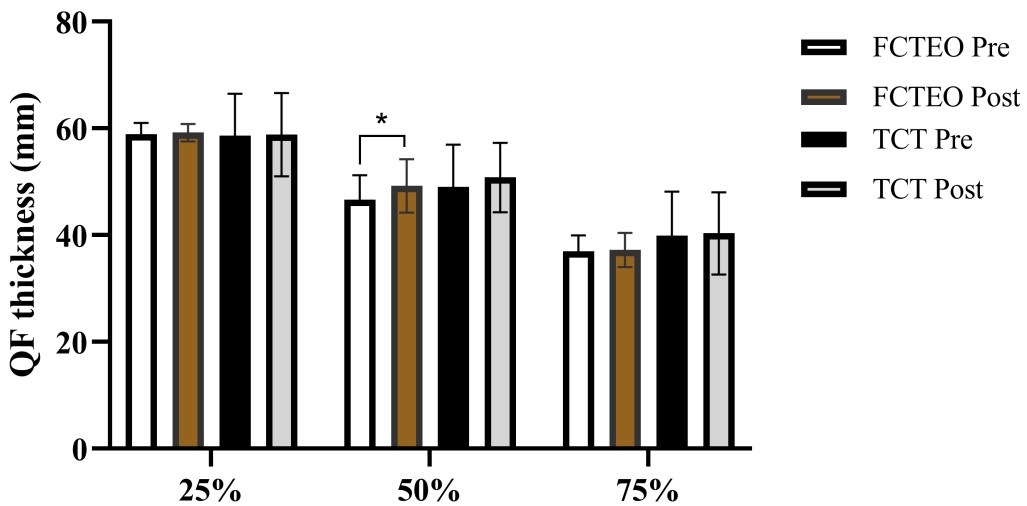

**Figure 4** QF thickness at proximal (25%), medial (50%), and distal (75%) measurements pre-test and post-test for both, FCTEO and TCT groups.

= 0.015, $P = 0.0908$, ES = 0.002) on QF25 were not significant. The interaction effect for MT at 50% of the QF length (QF50) between group and time was also not significant ($F\_group \times time = 0.068$, $P = 0.802$, ES = 0.011), nor was the main effect of group ($F\_group = 1.145$, $P = 0.326$, ES = 0.160). However, the main effect of time on QF50 was significant ($F\_time = 9.145$, $P = 0.023$, ES = 0.604). *Post-hoc* analysis revealed that the FCTEO participants significantly increased their MT at QF50 from pre- to post-intervention ($F\_FCTEO\ pre\ vs\ post = 6.103$, $P = 0.048$, ES = 0.504), while the TCT participants did not show a significant change ($F\_TCT\ pre\ vs\ post = 0.804$, $P = 0.404$, ES = 0.118). However, there was no significant difference in the MT at QF50 between the FCTEO and TCT groups after the training period (Post: $F\_FCTEO\ vs\ TCT = 4.044$, $P = 0.091$, ES = 0.403). No significant main effects of group, time, or the interaction between group and time were observed for MT at 75% of the QF length (QF75) ($F\_group = 1.649$, $P = 0.246$, ES = 0.216; $F\_time = 0.035$, $P = 0.858$, ES = 0.006; $F\_group \times time = 0.001$, $P = 0.974$, ES = 0.000, respectively).

## Maximal half-squat strength

The results for maximal half-squat strength are presented in Fig. 5. A significant main effect was observed for time ($F\_time = 28.672$, $P = 0.002$, ES = 0.827) and for the interaction between group and time ($F\_group \times time = 6.080$, $P = 0.049$, ES = 0.503). However, no significant main effect was found for the group variable alone ($F\_group = 2.819$, $P = 0.144$, ES = 0.320). *Post-hoc* analysis revealed that both the FCTEO and TCT groups experienced a significant increase in 1RM (one-repetition maximum) half-squat values after the training period ($F\_FCTEO\ pre\ vs\ post = 45.950$, $P = 0.001$, ES = 0.885;
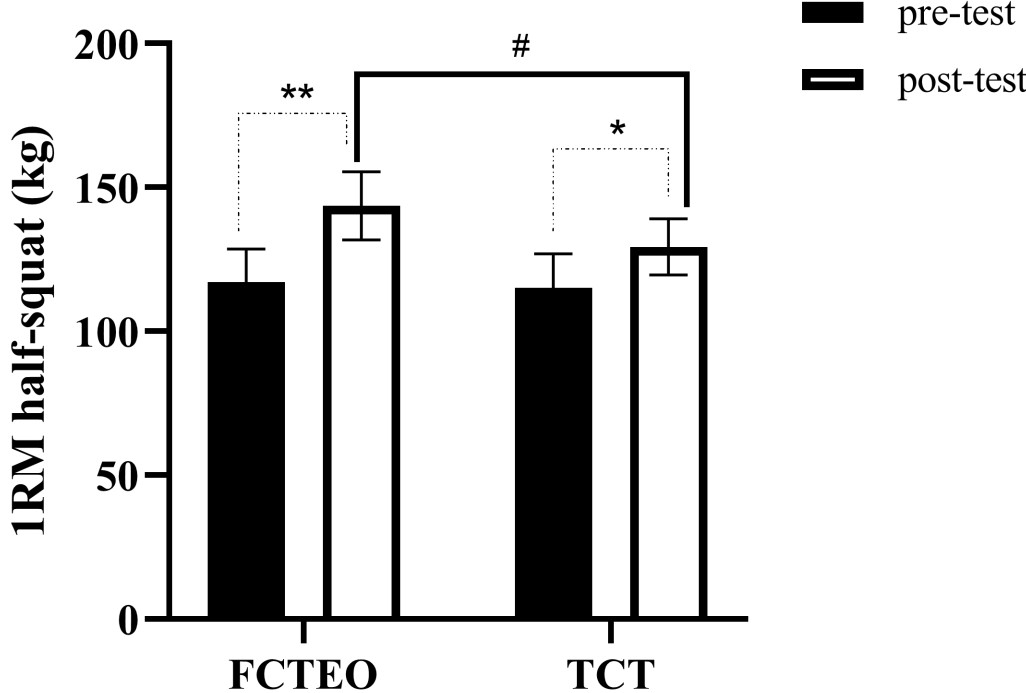

**Figure 5** **1RM half-squat measurements pre-test and post-test for both, FCTEO and TCT groups.**

F_TCT pre *vs* post = 7.886, $P = 0.031$, ES = 0.568, respectively). Furthermore, the *post-hoc* analysis indicated that the FCTEO group achieved a significantly higher 1RM half-squat value compared to the TCT group after the training period (Post: F_FCTEO *vs* TCT = 9.675, $P = 0.021$, ES = 0.671).

### Power
Vertical jump results are presented in Fig. 6.

### SJ
No significant main effects of group or group × time on squat jump (SJ) height were observed (F_group = 2.567, $P = 0.160$, ES = 0.300; F_group × time = 3.441, $P = 0.113$, ES = 0.364, respectively). However, a significant main effect of time on SJ height was noted (F_time = 10.451, $P = 0.018$, ES = 0.635). *Post-hoc* analysis revealed a significant increase in SJ height among TCT participants following the training intervention, compared to their baseline values (F_TCT pre *vs* post = 11.131, $P = 0.016$, ES = 0.650). Conversely, FCTEO participants did not exhibit a significant increase in SJ height post-training compared to baseline (F_FCTEO pre *vs* post = 1.749, $P = 0.234$, ES = 0.226). Furthermore, SJ height in the TCT group increased by 11%, while the FCTEO group experienced a 3% increase post-intervention.

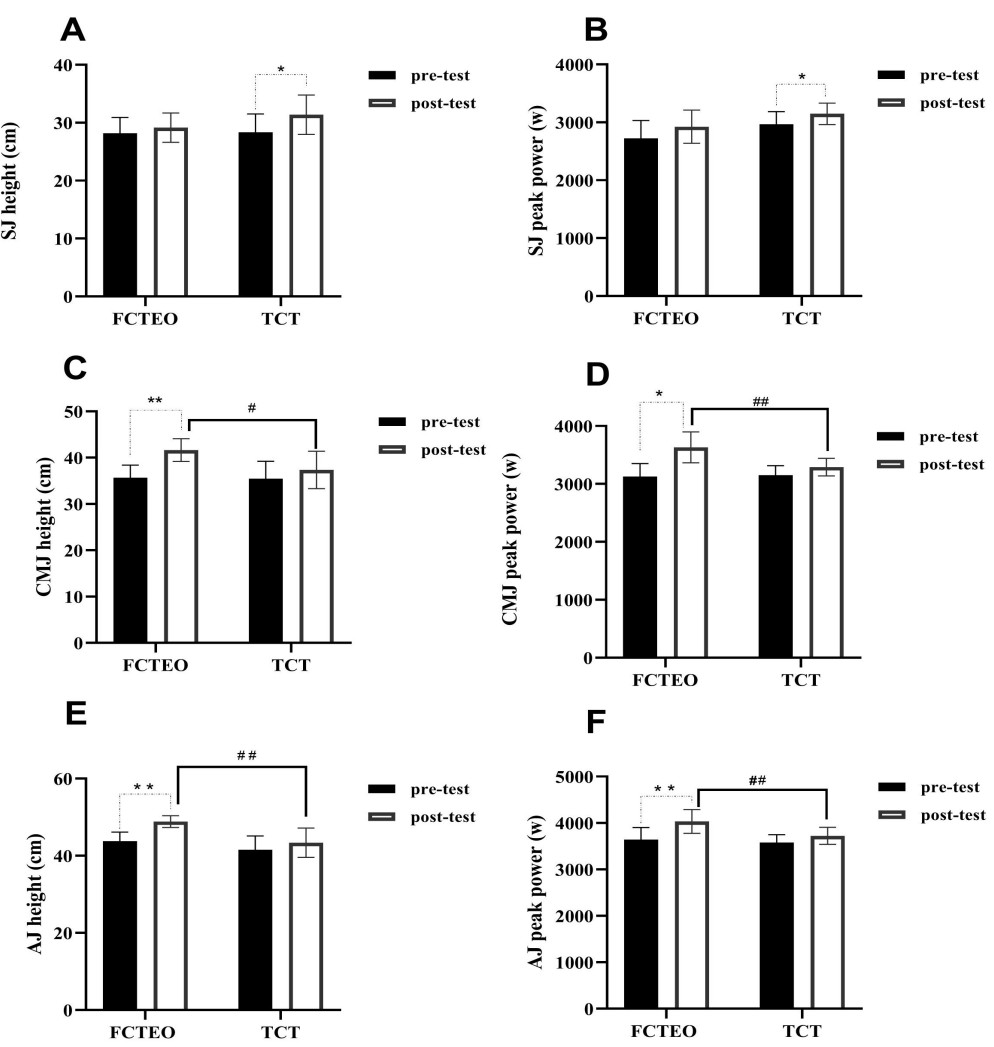

**Figure 6** (A–F) Power measurements pre-test and post-test for both, FCTEO and TCT groups.

Regarding SJ peak power, significant main effects of both group ($F\_group = 7.103$, $P = 0.037$, ES = 0.542) and time ($F\_time = 6.968$, $P = 0.039$, ES = 0.537) were found. *Post-hoc* analysis indicated a significant increase in SJ peak power for TCT participants after the training intervention compared to baseline ($F\_TCT$ pre *vs* post = 7.767, $P = 0.032$, ES = 0.564). In contrast, there was no significant increase in SJ peak power for FCTEO participants post-intervention compared to baseline ($F\_FCTEO$ pre *vs* post = 3.251, $P = 0.121$, ES = 0.351). Additionally, the peak power of SJ in the TCT group increased by 23%, whereas the FCTEO group showed a 7% increase after the training intervention.

Considering the effect sizes (ES) and the rate of change in SJ height and peak power post-intervention, it appears that TCT possesses greater potential for improving both SJ height and peak power in well-trained volleyball players.

## Countermovement jump (CMJ)

Significant main effects of group, time, and the interaction between group and time on countermovement jump (CMJ) height were observed (F_group = 6.547, $P = 0.043$, ES = 0.522; F_time = 18.968, $P = 0.005$, ES = 0.760; F_group × time = 7.242, $P = 0.036$, ES = 0.547, respectively). *Post-hoc* analysis demonstrated a significant increase in CMJ height for FCTEO participants post-training compared to baseline (F_FCTEO pre *vs* post = 15.140, $P = 0.008$, ES = 0.716). In contrast, the CMJ height of TCT participants did not show a significant increase post-training compared to baseline (F_TCT pre *vs* post = 1.563, $P = 0.258$, ES = 0.207). Additionally, *post-hoc* analysis revealed that FCTEO participants achieved significantly higher CMJ peak power than TCT participants post-training (Post: F_FCTEO *vs* TCT = 11.060, $P = 0.016$, ES = 0.648).

Significant main effects of time and group x time were also found for CMJ peak power (F_time = 9.161, $P = 0.023$, ES = 0.604; F_group × time = 7.597, $P = 0.033$, ES = 0.599, respectively), with FCTEO participants exhibiting significantly higher CMJ peak power than TCT participants after the training period (Post: F_FCTEO *vs* TCT = 17.952, $P = 0.005$, ES = 0.750). No significant main effect of group alone on CMJ peak power was observed.

Therefore, the results suggest that the FCTEO training program was more effective than TCT in enhancing CMJ height and peak power in well-trained volleyball players.

## Approach jump (AJ)

Significant main effects of group, time, and the interaction between group and time on approach jump (AJ) height were observed (F_group = 7.892, $P = 0.016$, ES = 0.397; F_time = 23.975, $P < 0.001$, ES = 0.666; F_group × time = 5.102, $P = 0.043$, ES = 0.298, respectively). *Post-hoc* analysis indicated a significant increase in AJ height for FCTEO participants post-training compared to baseline (F_FCTEO pre *vs* post = 25.599, $P < 0.001$, ES = 0.681). Conversely, there was no significant increase in AJ height for TCT participants post-training compared to baseline (F_TCT pre *vs* post = 3.479, $P = 0.087$, ES = 0.225). Furthermore, *post-hoc* analysis showed that FCTEO participants achieved significantly higher AJ height than TCT participants after the training period (Post: F_FCTEO *vs* TCT = 13.912, $P = 0.003$, ES = 0.537).

The study also identified a significant main effect of time and group x time on AJ peak power (F_time = 20.195, $P = 0.001$, ES = 0.627; F_group × time = 5.160, $P = 0.042$, ES = 0.301, respectively), while no significant main effect of group alone was found (F_group = 4.686, $P = 0.051$, ES = 0.281). *Post-hoc* analysis demonstrated a significant increase in AJ peak power for FCTEO participants post-training compared to baseline (F_FCTEO pre *vs* post = 22.885, $P < 0.001$, ES = 0.656). However, TCT participants did not show a significant increase in AJ peak power post-training compared to baseline (F_TCT pre *vs* post = 2.469, $P = .142$, ES = 0.171). Additionally, *post-hoc* analysis indicated that FCTEO participants had significantly higher AJ peak power than TCT participants after the training period (Post: F_FCTEO *vs* TCT = 9.470, $P = 0.010$, ES = 0.441).

Therefore, the results suggest that the FCTEO training program was more effective than TCT in enhancing both AJ height and peak power in well-trained volleyball players.

## DISCUSSION

According to the authors, this study is the first to compare the muscular adaptations in elite volleyball players undergoing FCTEO *versus* TCT. The findings reveal that, following an 8-week training program integrating flywheel complex training into a competitive volleyball training regimen, there were more significant increases in MT, strength, and performance in both the CMJ and the AJ compared to those achieved through traditional complex training. However, a notable improvement in squat jump (SJ) performance was exclusively observed in the group following TCT. This suggests that while FCTEO may be more effective in enhancing certain aspects of muscular development and jump performance, TCT still holds specific benefits, particularly for SJ performance.

### Muscle hypertrophy

Our results indicated that significant improvement in muscle thickness (MT) at the midpoint of the quadriceps femoris (50% QF) post-training was observed exclusively in the FCTEO group. Although no significant difference was found between the FCTEO and TCT groups, the increase in 50% QF MT in the FCTEO group (6.0%) was twice that observed in the TCT group (3.0%), aligning with findings from previous studies. *Norrbrand et al. (2008)* reported a 6.2% increase in quadriceps muscle volume following five weeks of flywheel resistance training (FRT), which was double that seen in traditional resistance training (3.0%), although no significant intergroup difference was noted. Similar increaseshave been observed in healthy men and women following FRT, such as a 5% increase in leg muscle mass (*Fernandez-Gonzalo et al., 2014*), a 7% increase in CSA (cross-sectional area) (*Seynnes, De Boer & Narici, 2007*), and an 11.4% increase in rectus femoris MT (*Horwath et al., 2019*). The enhancement of rectus femoris hypertrophy through flywheel half-squats training may be partly attributable to the greater activation of the rectus femoris during this training modality (*Illera-Domínguez et al., 2018*; *Kubo, Ikebukuro & Yata, 2019*). Studies have demonstrated a greater exercise-induced contrast shift and increased transverse relaxation time (T2) in magnetic resonance imaging (MRI), a measure of muscle usage during exercise and a predictor of CSA increase, following flywheel half-squats compared to barbell half-squats. Moreover, T2 readings indicated that rectus femoris usage increased more with flywheel half-squats ($+24 \pm 14\%$) than with barbell half-squats ($+8 \pm 4\%$) (*Norrbrand et al., 2011*). On the other hand, there is evidence suggesting that eccentric activities result in greater myofibrillar disruption and muscle damage compared to concentric actions (*Friden & Lieber, 1992*; *Gibala et al., 1995*). Previous studies have established that muscle injury induced by such loading is a pivotal stimulus for myofibrillar remodeling and subsequent muscle growth (*Evans & Cannon, 1991*; *Yu, Fürst & Thornell, 2003*). These findings suggest that specific exercises, such as half-squats that emphasize eccentric actions, could lead to more pronounced improvements in rectus femoris hypertrophy compared to traditional resistance training methods.

## Maximal muscle strength

The present results demonstrated that FCTEO training was more effective in improving strength in the half-squat compared to TCT training, showing an increase of 22.5% in the FCTEO group and 12.4% in the TCT group ($p = 0.021$, ES = 0.671). This finding aligns with the results of *Fernandez-Gonzalo et al. (2014)* who observed a 25% and 20% increase in 1RM for men and women, respectively, after 6 weeks of training on a flywheel squat device, albeit without a control group. In contrast to our results, *Maroto-Izquierdo, García-López & De Paz (2017a)* found no significant differences in the increase of the 1RM leg press between FRT and TRT programs in well-trained handball players. This discrepancy could be attributed to differences in training duration and session frequency. Longer training durations and more frequent sessions have been shown to significantly impact resistance training adaptations (*Moran et al., 2017*). Our study's intervention lasted 8 weeks, encompassing 24 sessions in total, whereas the intervention of *Maroto-Izquierdo, García-López & De Paz (2017a)* was only 6 weeks long with 15 sessions. Hence, FCTEO training appears more effective in enhancing strength compared to TCT, with training duration and frequency potentially playing a significant role in the effectiveness of different FCTEO training methods.

Furthermore, the FRT used in the FCTEO group in our study differed substantially from the TRT in the TCT group. During flywheel exercises, the rotational inertia generates a greater eccentric overload compared to TRT (*Maroto-Izquierdo et al., 2017b*).The flywheel provided unrestricted resistance at all intensities and maximal or near-maximal activation from the outset due to its inertia force (*McErlain-Naylor & Beato, 2021*). In contrast, maximal muscle activation during TRT typically occurs at contraction failure or ''sticking point'', leading to non-maximal forces at the beginning of a set and a decline in force throughout the set (*Norrbrand, Pozzo & Tesch, 2010*). Exercise intensity is a key determinant of strength training-induced adaptations (*Campos et al., 2002*; *Heggelund et al., 2013*). Additionally, eccentric loading led to specific neuromuscular changes such as attenuated motor recruitment (*Douglas et al., 2017*), preferential recruitment of high threshold motor units, and increased cortical activity (*Hody et al., 2019*). Furthermore, an increase in eccentric phase force production has been linked to enhanced concentric phase force output (*Doan et al., 2002*; *Takarada et al., 1997*). Collectively, these physiological differences support our study's findings and highlight the efficacy of flywheel training in improving the strength of female volleyball players.

## Muscle power

The results of CMJ in this study align with those from previous research. Studies involving elite soccer and basketball players have demonstrated that FRT leads to a significant increase in CMJ height (*Seynnes, De Boer & Narici, 2007*; *Stojanović et al., 2021*). Conversely, research involving physical education students did not report notable increases in SJ or CMJ performance, which might be attributed to the elite status of our study's participants. Indeed, factors such as training age (*Till et al., 2017*), and strength level (*Prue, McGuigan & Newton, 2010*) have been shown to influence changes in power following resistance training. However, FCTEO participants did not exhibit a significant

effect on SJ performance, potentially due to the flywheels less intense stimulation of concentric muscle contraction compared to TCT (*MacDougall, MacDougall & Sale, 2014*).

The substantial improvements in power tests (CMJ and AJ) in the FCTEO training group may be attributed to the following reasons: (1) Myogenic factors, which include increased half-squat strength that directly relates to the CMJ test technique, making the transfer of strength gains from FCTEO training to CMJ relatively straightforward. PAPE also plays a critical role, as positive PAPE effects on jumping performance following FRT have been reported (*Beato et al., 2019*; *Beato et al., 2021a*; *Beato et al., 2021b*; *Maroto-Izquierdo, Bautista & Rivera, 2020*; *Timon et al., 2019*). Additionally, increased muscle–tendon stiffness has been observed, *Onambélé et al. (2008)* reported a 136% increase in gastrocnemius lateralis and soleus tendon stiffness after 12 weeks of FRT. (2) Neuronal factors, including improvements in the stretch-shortening cycle (SSC) (*Bosco, Komi & Ito, 1981*), an increase in the excitability threshold of the Golgi tendon organs (*McNeely & Sandler, 2006*), and changes in neuromuscular coordination, which have been noted to influence jumping performance. Previous studies have shown that neuromuscular control improves during inertial resistance training (*Seynnes, De Boer & Narici, 2007*; *Tesch et al., 2004*). Consequently, it can be inferred that the enhancement of explosive power performance through FRT may be due to both myogenic and neuronal factors.

Certain limitations of this study warrant additional consideration. Given the specific competitive level of our participants, the results may only be generalizable to athletes of a similar competitive standard. Additionally, due to the off-season timing of the training intervention, this study did not explore the effects of FCTEO on field-based performance measures such as sprint speed and change of direction (COD) ability. However, including such measurements could have provided further insights into how FCTEO training transfers to team sport tasks, as previous research has shown that FRT results in greater improvements in speed and COD than TRT (*Maroto-Izquierdo et al., 2017b*). Our study focused exclusively on the lower body; hence, future research should also investigate the effects of FCTEO on upper body performance. This is particularly relevant for volleyball players, where upper body involvement is critical in training and competition activities like spiking, serving, and blocking (*Sheppard et al., 2007*). Despite these limitations, our study reported greater efficacy for FCTEO training. It is recommended that future research explore the use of surface electromyography in the application of FCTEO to gain deeper insights into muscle activation and training effects.

## CONCLUSIONS

FCTEO training has been shown to be more effective in improving muscular adaptation in elite female volleyball players. These findings carry significant implications for professionals working with this demographic. Considering the critical importance of enhancing muscle hypertrophy, strength, and power in female volleyball players, the training interventions discussed in this study offer practical guidance on implementing the FCTEO approach to cultivate these attributes. Future research examining FCTEO in athletes should explore the impact of various training prescription variables, such as

intensity, volume, and frequency, on optimizing strength, power, and speed. Additionally, it is important to investigate the biomechanical and molecular biological mechanisms underlying the effects of FCTEO training to advance its scientific application in competitive sports.

### Funding
The authors received no funding for this work.

### Competing Interests
The authors declare there are no competing interests.

### Author Contributions
- Jiaoqin Wang conceived and designed the experiments, performed the experiments, analyzed the data, prepared figures and/or tables, authored or reviewed drafts of the article, and approved the final draft.
- Qiang Zhang conceived and designed the experiments, performed the experiments, analyzed the data, prepared figures and/or tables, and approved the final draft.
- Wenhui Chen performed the experiments, prepared figures and/or tables, and approved the final draft.
- Honghao Fu performed the experiments, prepared figures and/or tables, and approved the final draft.
- Ming Zhang conceived and designed the experiments, analyzed the data, authored or reviewed drafts of the article, and approved the final draft.
- Yongzhao Fan conceived and designed the experiments, analyzed the data, authored or reviewed drafts of the article, and approved the final draft.

### Human Ethics
The following information was supplied relating to ethical approvals (*i.e.*, approving body and any reference numbers):

Each individual gave written agreement to participate in this study, which was authorized by the Capital University of Physical Education and Sports's Ethics Council2021A39.

### Ethics
The following information was supplied relating to ethical approvals (*i.e.*, approving body and any reference numbers):

Capital University of Physical Education and Sports's Ethics Council (2021A39).

### Data Availability
The raw data are available in the Supplementary File.

## Supplemental Information

Supplemental information for this article can be found online at http://dx.doi.org/10.7717/peerj.17079#supplemental-information.

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
