# Peer review of "The effect of flywheel complex training with eccentric-overload on muscular adaptation in elite female volleyball players"

_PeerJ, doi:10.7717/peerj.17079_

## Round 0.1 · original submission · Major Revisions

Dear Authors

Three experts in the study field have reviewed the manuscript. Your study provides valuable input for our current understanding of applied sciences and volleyball performance. The reviewers have addressed several issues for the next round of revision. We invite you to submit a revised version of the manuscript that addresses the points raised by the reviewers, particularly for the comments raised by review 1.

We look forward to receiving your revised manuscript.

Best regards

Yung-Sheng Chen, Ph.D.
Academic Editor

**Language Note:** The review process has identified that the English language must be improved. PeerJ can provide language editing services - please contact us at copyediting@peerj.com for pricing (be sure to provide your manuscript number and title). Alternatively, you should make your own arrangements to improve the language quality and provide details in your response letter. – PeerJ Staff

Reviewer 1 ·

Basic reporting

Please, see point "4. Additional comments" for the complete review points.

Experimental design

Please, see point "4. Additional comments" for the complete review points.

Validity of the findings

Please, see point "4. Additional comments" for the complete review points.

Additional comments

Manuscript: #87748
GENERAL COMMENTS: The present study covered a topic widely researched in recent years but the authors tried to explain how their study differed from the others recently published. The authors should be acknowledged for their work since conducting research in real world settings is challenging and not always possible. However, I have some major issues with some of the interpretations made by the authors when it comes to the analysis performed. In multiple occasions, the authors concluded that one intervention was superior to the other when no group x time interaction was found. They state that one group outperformed the other based solely on group main effects or time main effects which led the conclusions that may be not accurate and that may not help coaches and practitioners to make better informed decisions with their athletes (which should be the end goal of any study). Also, there are many grammatical flaws within the text that make it really diffic qult to understand at times. In light of these fatal flaws, I have no option but to recommend the rejection of the study. Nevertheless, I present several specific comments below that I hope can help the authors. Please, see below:

Abstract
Line 9. Please, replace "semi-squats" with "half-squats" throughout the whole document. This term is much more used in the scientific literature.

Line 11. Consider replacing "results which were measured" with "variables assessed".

Introduction
Line 24-26. "In competitive team sports such as basketball, volleyball, and American football, muscle strength and power are essential because they form the basis for performance-determining activities such as jumping, running, and hitting." Please include references to support these claims.

Line 38-40. "Complex training (CT), a form of combination training, is best described as training that alternates between traditional RT (heavy RT) and PT (light RT) within a single training session (Chu, 1996; Ebben and Blackard, 1997; Ebben, 2002)."
Due to conflicting and confusing terminology when it comes to Complex Training a recent study has proposed a standardized terminology. The authors are referred to the following study for more information on this and should use the appropriate terminology:
Cormier, P., Freitas, T. T., Loturco, I., Turner, A., Virgile, A., Haff, G. G., ... & Bishop, C. (2022). Within session exercise sequencing during programming for complex training: historical perspectives, terminology, and training considerations. Sports Medicine, 52(10), 2371-2389.
Line 41-48. "Nonetheless, in traditional complex (...) contraction (Alkner et al., 2003)".
This sentence is too long and difficult to follow. Please rephrase it to make it easier for the reader to understand.

Line 55. What do the authors mean with "flywheel paradigm"?

Line 62. "post-activation performance". Do the authors mean "post-activation performance enhancement"

Methods

Line 80. Consider replacing "the training interventions lasted three times" with "the training sessions were performed three times"... Also, what do the authors mean by "muscle viscoelastic point"? Please, clarify.

Line 85. The sentence should start with: "Fourteen women from a high-level volleyball team, whose competitive level was Division I, were selected for the study".

Line 103. "A one-week test was followed by a one-week familiarization session."
This is not clear. The familiarization was completed after the testing? How can the authors ensure that the testing procedures were correctly performed by the players? Also, what do you mean by a "one-week familiarization session"? It lasted the whole week? This study design is highly confusing.

Line 104. Replace "participants was" with "participants were"

Line 108-109. "Wednesday, 1RM parallel semi-squat measure."
This sentence is not grammatically sound. Please, correct.


Line 116-117. "Weekend, muscular power tests include SJ, CMJ, and three-step approach AJ".
Again, this sentence is not grammatically sound. Please, correct.

Lin 119. Did all players perform the progressive loading test on the flywheel or only the ones that were assigned to the flywheel group? This is not clear for the reader.

Line 124-126. "A washout interval of 48 to 72 hours was permitted. Participants conducted the first experimental session (FCTEO or TCT) at the same time each day, followed by the second experimental session."
This is not clear for the reader. The washout period was between what and what? Also, what do the authors mean by first and second experimental session? The way it is written it appears that all participants performed both training programs as in a crossover design study. Please re-write to avoid confusion,

Line 128. Replace "Hypertrophy Testing" with "Muscle Thickness Assessment"

Line 132. Please insert "imaging" after "Ultrasonic"

Line 147. Replace "Power Testing" with "Vertical Jump Testing"

Line 156-157. "In all jumping tests, participants had to land in the same location from which they had taken off to avoid bending their knees and varying their measurements (Markovic et al., 2004)."
Why would landing on in the same location avoid players from bending their knees? This sentence is not clear. Please, explain.

Line 159-160. "with the best jumping recorded for analysis."
Best jump based on what? Jump height? And which metrics were assessed? Jump height? Peak Power? Also, is smart Jump a force platform or a contact mat? Please, clarify and also indicate how power was calculated/obtained.

Line 172. "Experimental Procedures". the authors should replace with "Training Interventions" or "Training Programs".

Line 189. Replace "each repeat" with "each repetition".

Line 192. "The participants were asked to rate their felt exertion at the end of each set."
This sentence is not grammatically correct. Please, re-write.

Results
Line 224-226. "However, the main effect of time on QF50 was significant (Ftime = 9.145, P = .023, ES = .604), indicating that only the FCTEO participants significantly increased their muscle thickness of QF50 after the training period (p = .048, ES = .504)."
This is not clear for the reader. How can the authors conclude that only the FCTEO group increased significantly muscle thickness here? The main effects of time only indicate that there were pre-post differences, irrespective of the group (i.e., when the sample is considered as a whole, FCTEO + TCT combined). Thus, this conclusion cannot be obtained with the statistical analysis results. Please, clarify.

Figure 4. This figure would be much improved if the authors designed it differently. I recommend they place the groups in the horizontal axis with the columns corresponding to the pre and post-test.

Line 246. "Power results". Replace with "vertical jump results" since power was not the only variable assessed.

Figure 5. Same as previous comment regarding Figure 4.

Line 249-252. "However, the main effect of time had a significant main effect on SJ height (Ftime= 10.451, P = .018, ES = .635). Post-hoc analysis indicated that TCT participants significantly increased SJ height compared to FCTEO participants after the training period"
If there is no group x time interaction, how can the authors conclude that one group is superior to the other? I am afraid that is not correct.

Line 253-256. Same as the above comment.

Line 258-261. "Significant main effects of group and time on CMJ height were found (Fgroup = 6.547, P = .043, ES = .522, and Ftime = 18.968, P = .005, ES = .760, respectively). Post-hoc analysis showed that FCTEO participants had significantly higher CMJ height than TCT participants (FFCTEO = 11.060, P = .016, ES = .648) after the vs. TCT training period."
Again, this conclusion is difficult to understand. If there is no group x time interaction it is not possible to conclude that one intervention was superior to the other.

Line 268-277. The results section of the "AJ" variable is questionable as well. The authors make claims that are not supported by the results of their analysis.

Discussion
Overall, the discussion is confusing and the text is hard to follow (missing words, periods when there should be commas, sentences with no meaning, etc). Moreover, the authors discuss the results and make conclusions that are quite an overreach based on the misinterpretation of some of the parameters of the statistical analysis. Therefore, the current version of the discussion is inaccurate and may lead the reader and the end user (i.e., practitioners on the field) into erroneous conclusions. This whole section would need to be re-written from scratch once the results are properly described and interpreted.

Reviewer 2 ·

Basic reporting

This is a study on the impact of eccentric-overload flywheel compound exercises and traditional Smith machine compound resistance exercises on the lower limb muscle thickness, maximal strength, and explosive power of female elite volleyball players from the perspective of sports science. This is indeed an intriguing topic, as research comparing different training modalities to enhance the specific qualities of female elite athletes is particularly important. The article demonstrates that the eccentric-overload flywheel compound exercise modality can enhance the lower limb muscle adaptation of female elite volleyball players, as indicated by measurements of quadriceps thickness, maximal half-squat strength, and squat jump performance. However, the paper needs very significant improvement before acceptance for publication. The following is my comments and critique.

Experimental design

Strengths:
This article represents the first-ever comparison of eccentric-overload flywheel resistance exercises and traditional compound resistance exercises in terms of their impact on muscle adaptation in female elite volleyball players.

Limitations:
Unfortunately, the authors only focused on the maximal strength, explosive power, and muscle thickness of the lower limb skeletal muscles of the athletes, without conducting a detailed comparison of muscle strength in various regions (upper limbs, trunk). Moreover, the whole manuscript must be edited by a fluent English speaker.

Validity of the findings

Abstract

Line 10:” The results” should be changed to” The indicators”.

Line 20: It is recommended to add “muscular adaptation” in Keywords.

Introduction

Line 28: It is recommended to delete “The strength and power of an athlete's lower limbs account for the majority of their jumping capacity”.

Line 65-70: There is a logical error in the entire paragraph. The indicators such as squat jump (SJ), counter movement jump (CMJ), and three step approach jump (AJ) should be placed before this paragraph.

Methods

Line 87: "Fourteen women were assigned randomly to either the FCTEO or TCT group." should be deleted as it is repeat with the previous content.

Line 129/131: the “(MT)” should be removed since “muscle thickness” has already been abbreviated in the previous context.

Suggest merging “Measures” with “Experimental Procedures” and separately listing the steps for the eccentric-overload flywheel compound exercises.

Results

Picture 4: please enlarge the group identifier in the upper right corner, as it is not clear.

The text mentions monitoring the athletes “levels of perceived effort” during the exercise, but there is no information regarding the results. Please add.

Discussion

Line 317:”since” should be change to “Since”.

Conclusions

“Despite the fact that our findings suggested that FCTEO may be superior to TCT, FCTEO is a good training strategy for enhancing lower-body muscular thickness, strength, and power”. Logic error, please amend.

Additional comments

no additional comment.

Annotated reviews are not available for download in order to protect the identity of reviewers who chose to remain anonymous.

·

Basic reporting

The manuscript requires clarity in presenting the broader problem in the opening paragraph.
There is an inconsistency in the usage of the term 'power.' It should be consistently referred to as 'power output.'
Acronyms and abbreviations need proper introduction before usage throughout the document.
Ensure that the manuscript is free from redundancies and repetition of concepts. Cohesiveness and flow between sections must be ensured, especially when referencing hypertrophy.
Figures should be modified to enhance transparency. While bar graphs are used, individual data points are not discernible, making it difficult to visualize the data distribution.

Experimental design

The chosen variables, especially those related to jump height and plyometrics, require a clear definition and justification.
The rationale behind using a specific training intensity (like 80%) should be justified.
The usage of maximum concentric speed needs clear reasoning.
There is ambiguity regarding the chosen heights for plyometric exercises, and how they were individually determined needs explanation.
The categorization of plyometrics as low intensity requires clarification, given the definition provided for Complex Training.

Validity of the findings

The reliability of the chosen evaluation method should be stated, preferably with objective indicators like CV%.
Linking findings to the broader problem posed at the beginning will enhance the paper's validity.
Statements, such as the impact of hypertrophy, need to be backed by a prior introduction or justification in the relevant sections.

Additional comments

Some paragraphs lack a clear conclusion or closing statement, affecting the overall structure and understanding of the content.
Certain content might be better suited in a limitations section or removed if it does not contribute to the overall study's rationale.
The term 'semi-muscle strength' is ambiguous and requires clarity.

---

## Round 0.2 · accepted · Accept

Dear Authors,

I would like to express my thanks for your patience and efforts to improve the quality of the manuscript. Your submission is now endorsed by three experts for acceptance of publication in PeerJ. Congratulation!!!

Thank you for submitting your interesting article to PeerJ. I look forward to receiving your future research and review articles for consideration.

Best Regards
Ph.D. Yung-Sheng Chen

Reviewer 1 ·

Basic reporting

The authors have made a thorough revision of the results section of the manuscript. The issues initially raised have been addressed and corrected, thus leading to conclusions that are supported by the statistical analysis. I consider that the manuscript has greatly improved and, according to the other reviewer comments, I will not oppose the publication of the study. I support its acceptance. Congratulations!

Experimental design

No comment

Validity of the findings

The authors have made a thorough revision of the results section of the manuscript. The issues initially raised have been addressed and corrected, thus leading to conclusions that are supported by the statistical analysis. I consider that the manuscript has greatly improved and, according to the other reviewer comments, I will not oppose the publication of the study. I support its acceptance. Congratulations!

Reviewer 2 ·

Basic reporting

The authors have addressed all my concerns, so I would like to recommend the paper for publication.

Experimental design

I have no further comments

Validity of the findings

I have no further comments

·

Basic reporting

No comments

Experimental design

No comments

Validity of the findings

No comments

Additional comments

I want to extend my sincere gratitude for the author's thorough and thoughtful responses to each of the queries I raised. Your dedication to engaging with the review process has significantly contributed to enhancing the quality and clarity of your manuscript. It is commendable to see authors who are so committed to the rigors of scientific discourse and peer review.